# Modelling the influence of meteoric smoke particles on artificial heating in the D-region

Margaretha Myrvang[1], Carsten Baumann[2], and Ingrid Mann[1]

[1]UiT The Arctic University of Norway, Department of Physics and Technology, Postboks 6050 Langnes, 9037 Tromsø
[2]German Aerospace Center, Institute for solar-Terrestrial Physics, 17235 Neustrelitz, Germany

**Correspondence:** Margaretha Myrvang (margaretha.myrvang@uit.no)

**Abstract.** We investigate if the presence of meteoric smoke particles (MSP) influences the electron temperature during artificial heating in the D-region. By transferring the energy of powerful high frequency radio waves into thermal energy of electrons, artificial heating increases the electron temperature. Artificial heating depends on the height variation of electron density. The presence of MSPs can influence the electron density through charging of MSPs by electrons, which can reduce the number of free electrons and even result in height regions with strongly reduced electron density, so-called electron bite-outs. We simulate the influence of the artificial heating by calculating the intensity of the upward propagating radio wave. The electron temperature at each height is derived from the balance of radio wave absorption and cooling through elastic and inelastic collisions with neutral species.

The influence of MSPs is investigated by including results from a one-dimensional height-dependent ionospheric model that includes electrons, positively and negatively charged ions, neutral MSP, singly positively and singly negatively charged MSP and photo chemistry such as photo ionization and photo detachment. We apply typical ionospheric conditions and find that MSPs can influence both the magnitude and the height profile of the heated electron temperature above 80 km, however this depends on ionospheric conditions. During night, the presence of MSPs leads to more efficient heating, and thus a higher electron temperature, above altitudes of 80 km. We found differences of up to 1000 K in electron temperature for calculations with and without MSPs. When MSPs are present, the heated electron temperature decreases more slowly. The presence of MSPs does not much affect the heating below 80 km for night conditions. For day conditions, the difference between the heated electron temperature with MSPs and without MSPs is less than 25 K.

We also investigate model runs using MSP number density profiles for autumn, summer and winter. The night-time electron temperature is expected to be 280 K hotter in autumn than during winter conditions, while the sunlit D-region is 8 K cooler for autumn MSP conditions than for the summer case, depending on altitude. Finally, an investigation of the electron attachment efficiency to MSPs shows a significant impact on the amount of chargeable dust and consequently on the electron temperature.

## 1 Introduction

Meteoric smoke particles (MSP) are small nanometer-sized dust particles (Hunten et al., 1980; Plane, 2012). They can change the D-region charge balance by influencing the chemical processes through charging of MSP by electrons and ions [cf. Bau-

mann et al. (2015)]. By changing the charge balance, MSPs can influence artificial heating. The overall charge balance in the D-region is complex with positive ions, negative ions and cluster ions (Verronen et al., 2016).

The MSPs form as a result of meteor ablation that deposits the meteoric material in the higher atmosphere, which condense to MSPs of sizes up to a few nanometers (Plane, 2004). Measurements on-board rockets have detected both negatively and positively charged MSPs, indicating that MSPs can influence plasma densities in the D-region through charging of MSPs by electrons and ions (Friedrich et al., 2012). Charging of MSPs influences the charge balance mainly through electron attachment to MSPs, which can results in height regions with reduced electron density, so-called electron bite-outs. Electron bite-outs change the height profile of the electron density and this reduction in electron density occurs in altitude regions where the MSPs are most abundant. Electron bite-outs within the height profile of the electron density can affect the electron temperature during artificial heating, as shown by Kassa et al. (2005).

A heater transmits powerful high-frequency radio waves into the ionosphere during artificial heating experiments. In the collisional plasma of the ionospheric D-region, electrons absorb the radio wave energy transmitted from the heater and heat up, increasing the electron temperature. Consequently, the intensity of the radio wave decreases with height (Rietveld et al., 1986; Belova et al., 1995; Kero et al., 2000, 2008). Artificial heating can also induce different phenomena in the ionosphere, like for instance ion upwelling (Kosch et al., 2010) and artificial optical emission (Kosch et al., 2000, 2014) in the F-region. In the D-region, researchers have found that artificial heating can influence Polar Mesospheric Summer Echoes (PMSE) (Chilson et al., 2000; Havnes et al., 2004; Biebricher and Havnes, 2012). The PMSE form in the presence of atmospheric turbulence and charged ice particles and it is assumed that the presence of MSPs in the D-region facilitates the formation of ice particles.

The aim of our study is to numerically model the electron temperature during artificial heating and include the height variation of electron bite-outs by using an ionospheric model (Baumann et al., 2013) with MSPs. As a comparison, we also model without MSPs. The one-dimensional height-dependent ionospheric model is for quiet ionospheric conditions and includes MSPs and photochemistry such as photoionization and photodetachment. We calculate the artificial heating with different radio wave frequencies and higher or lower radio wave power to investigate if this influences the electron temperature and to check the robustness of our results. We will compare night and day conditions to see if a higher electron density during daytime influences the modelled electron temperature. In addition, we investigate the seasonal variation of the MSPs abundance, as well as the role of the electron attachment efficiency to MSPs for the heated electron temperature.

This paper is organized as follows. In part 2 we present a detailed theoretical background and numerical modelling of artificial heating in the D-region. Part 3 gives a brief description of the ionospheric model. In part 4 we introduce the results. Part 5 presents the discussion.

## 2 Artificial heating in the D-region

Powerful high frequency radio waves can heat up electrons in the ionospheric D-region by artificial heating experiments. The higher temperature of the electrons can lead to various phenomena in the whole ionosphere (e.g. Robinson (1989) and references therein). Artificial heating increases the electron temperature by transferring the radio wave energy into thermal

energy of electrons (Rietveld et al., 1986; Kero et al., 2007, 2008). Modelling of artificial heating in D-region altitudes shows an increase in electron temperature of a factor of 10 (Belova et al., 1995; Kero et al., 2000). The EISCAT high power high frequency heating facility located in Tromsø, Norway transmits powerful high-frequency radio waves into the ionosphere during artificial heating experiments. The ESICAT radar, also located in Tromsø, Norway, can observe the ionosphere during these heating experiments. The EISCAT heating facility in Tromsø has three different antenna arrays consisting of crossed full-wave dipoles with a frequency range of 3.85-8 MHz. There are 12 transmitters that can adjust the power output from 200 MW to 1200 MW, depending on the used radio frequency. The dipoles can transmit ordinary (O) circular polarization mode or extraordinary (X) circular polarization mode (Rietveld et al., 2016). The following model of the heated ionosphere, described in the next section, uses these experimental parameters of the EISCAT heating facility (Rietveld et al., 1993).

## 2.1 Description of model

This section describes the physical background of the artificial electron heating. For the implementation of the artificial electron heating, we rely on earlier work done by Rietveld et al. (1986); Belova et al. (1995); Kero et al. (2000); Kassa et al. (2005); Kero et al. (2007). Note that the model described in this section only covers the lower ionosphere. The heater transmits a powerful high frequency radio wave that propagates through the cold, magnetized, collisional plasma of the ionospheric D-region. The intensity *I*, or energy of the radio wave varies with height *h* according to:

$$\frac{dI}{dh} = -2kI \tag{1}$$

where *k* is the absorption coefficient, given as:

$$k = -\frac{\omega Im(n)}{c} \tag{2}$$

In Eq. 2, $\omega$ is the angular frequency of the heating radio wave, Im(n) is the imaginary part of the refractive index *n* and *c* is the speed of light. When integrated, Eq. 1 in combination with Eq. 2, yields the following expression for the intensity:

$$I(h) = \frac{ERP}{4\pi h^2} \exp\left(\frac{2\omega}{c}\int\limits_0^h Im(n)dh\right) \tag{3}$$

where ERP is the effective radiated power. For solving Eq. 3 we need an expression for the refractive index *n*. We can derive the refractive index by using the Appleton-Hartree dispersion relation, which describes the radio waves propagation in a cold magnetized plasma and which can be applied to the ionospheric D-region. It describes the refractive index as:

$$n^2 = 1 - \frac{X}{1 - iZ - \frac{(Y\sin\theta)^2}{2(1-X-iZ)} \pm \sqrt{\frac{(Y\sin\theta)^4}{4(1-X-iZ)^2} + (Y\cos\theta)^2}} \tag{4}$$

where $\theta$ is the angle between the wave vector and the direction of the magnetic field. Here, $(+)$ and $(-)$ represents the ordinary and extraordinary polarization modes, respectively. Note that the refractive index is complex $n = n_1 + in_2$. If the imaginary part is less than zero, the wave is damped. The wave damping is caused by wave energy loss through absorption by the plasma while the wave propagates through the ionosphere. Due to its lower mass, electrons absorb the energy and are thus heated, while ions and neutrals remain unheated in comparison. The dimensionless $X$, $Y$ and $Z$ are normalized frequencies, defined as:

$$X = \frac{\omega_{pe}^2}{\omega^2} = \frac{N_e e^2}{\varepsilon_0 m_e \omega^2} \tag{5}$$

$$Y = \frac{\omega_{ge}}{\omega} = \frac{eB}{m_e}\omega \tag{6}$$

$$Z = \frac{\nu_{en}}{\omega} \tag{7}$$

where $N_e$ is electron density, $e$ is unit charge, $\varepsilon_0$ is the permittivity of vacuum, $m_e$ is electron mass, $B$ is the Earth's magnetic field and $\nu_{en}$ is the electron-neutral collision frequency. How the electron-neutral collision frequency from Eq. 7 depends on electron temperature is taken from Dalgarno et al. (1967):

$$\nu_{en} = 1.7 \times 10^{-11}[N_2]T_e + 3.8 \times 10^{-10}[O_2]\sqrt{T_e} + 1.4 \times 10^{-10}[O]\sqrt{T_e} \tag{8}$$

where $[N_2]$ is the number density of molecular nitrogen, $[O_2]$ is the number density of molecular oxygen, $[O]$ is the number density of atomic oxygen and $T_e$ is the electron temperature. Neutral densities are in units of $\text{cm}^{-3}$ and temperature in K. Through $\nu_{en}$ the refractive index depends on the electron temperature. The electron-neutral collision frequency is high in the D-region due to the relatively low electron density in comparison to the neutral density. Therefore, electron ohmic heating is the dominant D-region ionospheric response to heating. In ohmic heating, electrons oscillating parallel to the radio wave electric field collide with neutrals. This causes a phase shift between the direction of the radio wave electric field and the direction of electron oscillation. Overall, electrons are scattered in a random direction. This random motion of electrons leads to absorption of wave energy, where the wave energy is transferred into thermal energy of electrons, increasing the electron temperature. To find the increased electron temperature we use the electron energy balance equation, which describes local electron energy conservation. Solving the electron energy equation gives us the electron temperature time variation due to energy input from the heater and cooling through collisions with neutrals. We have neglected thermal conductivity due to high neutral density in the D-region and neglected plasma transport. The electron energy equation is then given as:

$$\frac{dT_e}{dt} = \frac{2}{3k_b N_e}\left(Q(T_e) - L(T_e)\right) \tag{9}$$

where $k_b$ is Boltzmann's constant. Equation 9 is non-linear differential equation. Here $Q(T_e)$ is the power absorbed by electrons per volume:

$$Q(T_e) = 2k(T_e)I(h) = \frac{2\omega}{c} Im(n)I(h) \tag{10}$$

The electrons lose energy, and are thus cooled, through elastic and inelastic collisions with neutral species, where the inelastic collisions can excite vibrational and rotational states. The sum of all energy losses is given by the energy loss function $L(T_e)$; these are the electron cooling rates. Our cooling rates include vibrational and rotational excitation of molecular oxygen (Pavlov, 1998b) and of molecular nitrogen (Pavlov, 1998a), excitation of fine structure levels of atomic oxygen (Pavlov and Berringston, 1999) and elastic collisions between electrons and neutral species (Schunk and Nagy, 1978). Due to the low electron density in the D-region, we neglect electron-ion collision. More detailed descriptions of the electron cooling rates are in appendix B. When the heater is switched on, the electron temperature increases from its initial temperature, which is equal to the neutral temperature in the D-region, to a higher heated electron temperature. The heating time for this temperature increase is less than 1 ms due to the high collisions frequency $\nu_{en}$ in the D-region. After less than 1 ms the electron temperature has reached thermal equilibrium where $dT_e/dt = 0$. In cases where the heating modulation time is much longer than the heating time for the electron temperature, we can simplify Eq. 9 to:

$$Q(T_e) - L(T_e) = 0 \tag{11}$$

### 2.1.1 Implementation of model

To compute the electron temperatures during heating we numerically solve Eq. 11. At the first height the intensity is $I_0 = ERP/4\pi h^2$, the undamped radio wave. We then compute $Q(T_e)$ from Eq. 10 by using the intensity $I_0$, where $Q(T_e)$ is a function of $T_e$. We use the intensity $I_0$ to solve $Q(T_e) - L(T_e) = 0$ for the electron temperature by using an algorithm that combines the inverse quadratic interpolation method, bisection method and secant method (Brent, 1973; Forsythe et al., 1977). By solving $Q(T_e) - L(T_e) = 0$ we find the zero-point of Eq. 11, which gives us a new electron temperature. This new, modified electron temperature changes the refractive index since the electron-neutral collision frequency depends on electron temperature. With the changed refractive index, we recalculate the intensity, taking into account the loss due to absorption. We compute the intensity numerically by approximating the integral in Eq. 3 as a sum:

$$I(h) = \frac{ERP}{4\pi h^2} \exp\left( \frac{2\omega}{c} \sum_{h'=60km}^{h'=h} Im(n(h'))\Delta h \right) \tag{12}$$

where the first part $ERP/4\pi h^2$ represents the undamped radio wave and the part in the exponential function represents the damping effect due to absorption. The distance between each height is $\Delta h = (h') - (h' - 1)$. For our case $\Delta h = 1$ km and $\Delta h$ is constant for all altitudes. In the next iteration, the intensity has changed, so there is a new zero-point for $Q(T_e) - L(T_e) = 0$,

which we compute. Figure 1 shows $Q$ - $L$ as a function of $T_e$. This figure illustrate that loss due to absorption can change the location of the zero-point of $Q$ - $L$. We have used the following parameters to calculate $Q$ - $L$: Height 75 km, ionospheric night conditions, model run with the presence of MSPs, frequency 5 MHz and power 700 MW. Figure 1 shows the zero-point of $Q$ - $L$ with $I_0$, illustrated as a blue-coloured star and the zero-point of $Q$ - $L$ with the changed intensity $I_1$, illustrated as a magenta-coloured star. We see in Fig. 1 that the zero-point of $Q$ - $L$ is different for $I_0$ and $I_1$. This process with a new, modified electron temperature, which changes the intensity, is repeated in an iteration scheme. The neutral temperature is the starting point in the iteration scheme. The iteration scheme is repeated until the change in the electron temperature is very small, i.e. when $T_e$ converges. This equation visualizes the iteration for the intensity:

$$I(h+1) = I_0 - dI(h) - dI(h+1) \tag{13}$$

where $I_0 = ERP$. Here *dI(h)* represents absorption at heights below and *dI(h+1)* represents absorption at the current height. Before we move to the next height, we sum all the absorption, so that for the next height we take into account all absorption below. In the next height, we repeat the procedure described for the first height and calculate the heated electron temperature and the new intensity. This is done for all heights, moving upward from the initial height to the final height. Our altitude range is 60-120 km. The ionospheric D-region varies in altitude range from about 50 km to 100 km, however, we model up to 120 km to see if the electron temperature at altitudes above 100 km is influenced by the presence of MSPs at lower altitudes below.

We model the height-dependent heated electron temperature with initial height profiles for the following parameters: Earth's magnetic field, electron density, neutral temperature, neutral densities of molecular nitrogen, molecular oxygen and atomic oxygen. For Earth's magnetic field, we use a dipole approximation (Brekke, 2013) . The magnetic field goes into Eq. 6, which we use to compute the refractive index in Eq. 4. We compare day and night conditions to see if a higher ionization level, as during day condition, has an influence on the heated electron temperature. The used electron density height profiles during day and night conditions come from an ionospheric model by Baumann et al. (2013). The neutral temperature and neutral densities are from MSISE-90 model (Hedin, 1991; Picone et al., 2002) with the same date, time and location as used for the ionospheric model. The parameters for the EISCAT heating radio wave include polarization, radio wave frequency and effective radiated power (ERP). The model calculations are done with X-mode transmission polarization. For the radio wave frequency and ERP, we assume a number of different typical values to see if this influences the heated electron temperature with MSPs and without MSPs. We ran the model for four different cases, see Table 1 (Erik Varberg, personal communication). Figure 2 shows a schematic on how we computed the heated electron temperature by combining artificial heating and the electron density from the ionospheric model. In the next section we briefly describe the ionospheric model.

## 3  Background ionospheric model

Here we give a brief description of a one-dimensional height-dependent ionospheric model for the D-region, which includes MSPs, developed by Baumann et al. (2013). For the full description, see Baumann et al. (2013) and references therein. The

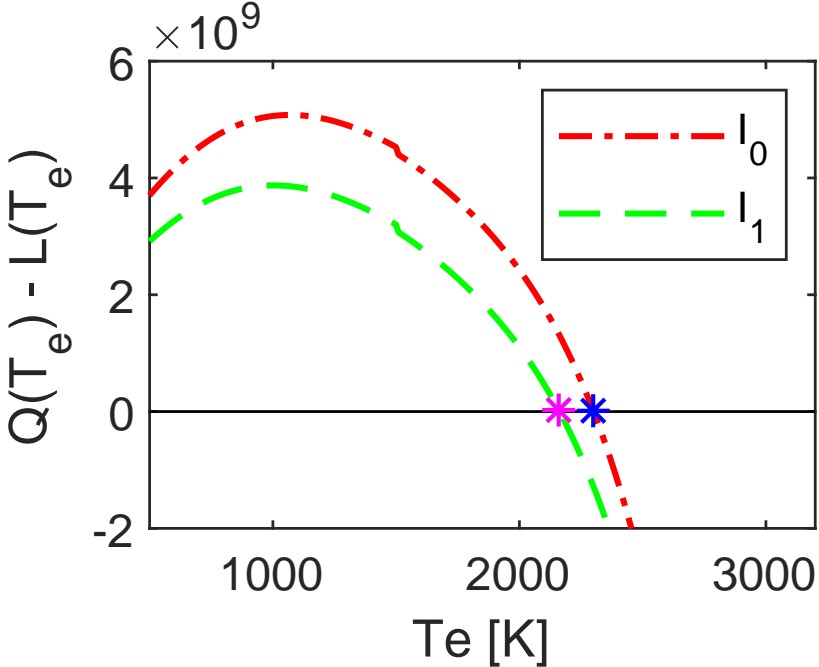

**Figure 1.** Illustration of $Q(T_e) - L(T_e)$ as a function of electron temperature with intensity $I_0$ (the undamped radio wave) and $I_1$ (radio wave with damping). Here $I_0 > I_1$. The units of $Q(T_e) - L(T_e)$ is energy per volume per second $\mathrm{eVm^{-3}s^{-1}}$. With a different intensity, we change the location of the zero-point, where $Q(T_e) - L(T_e) = 0$. The zero-point is marked as a blue star for $I_0$ or a magenta star for $I_1$. We have used the following parameters to calculate $Q$ - $L$: Height 75 km, ionospheric night conditions, model run with the presence of MSPs, frequency 5 MHz and power 700 MW.

**Table 1.** The frequencies and effective radiated power (ERP) used in the study.

| Case 1 | Case 2 | Case 3 | Case 4 |
|--------|--------|--------|--------|
| 4 MHz | 5.5 MHz | 5.5 MHz | 7.5 MHz |
| 200 MW | 300 MW | 600 MW | 1200 MW |

one-dimensional height-dependent ionospheric model is run for quiet ionospheric conditions between heights of 60-120 km and includes electron, positively and negatively charged ions, neutral MSP, singly positively and singly negatively charged

MSP. Multiply charged dust is unlikely to occur since the MSPs are very small. Initial conditions for the height and size-dependent MSP number density profile come from Megner et al. (2006). The size range is from 0.2 nm to 41 nm. Above 100 km, the number density of MSPs is assumed to be very small. Megner et al. (2006) calculates the MSP number density profile by using a one-dimensional model, where the MSP height distribution varies with size. The number density of smaller MSPs (less than 15 nm) increases with altitude, while larger sizes are more abundant at lower altitudes between 60-70 km.

Overall the number density of MSPs increases from 60 km to a maximum at around 80-83 km, and then decreases above. For

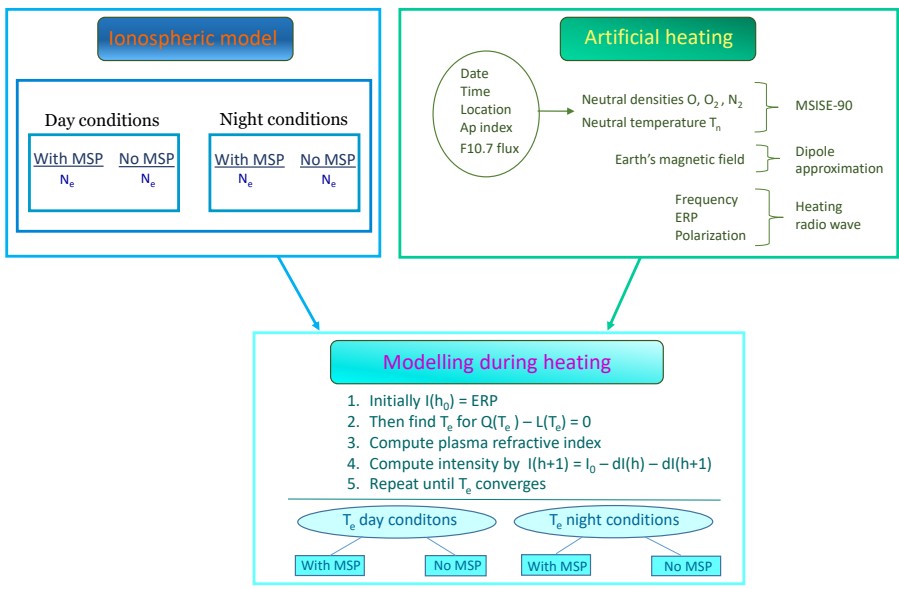

**Figure 2.** Schematic showing how we combined artificial heating and the electron density from the ionospheric model in order to compute the heated electron temperature. The parameters for artificial heating include frequency, effective radiated power (ERP) and polarization of the heating radio wave, Earth's magnetic field and neutral densities and neutral temperature. We perform the modelling during heating at each height from below before going to the next height, moving upward from the initial height to the final height.

an overview of the different MSP number density profiles, we refer the reader to figure A1 in the appendix. For the charging of a MSP by electrons, the electron attachment efficiency is the probability of a MSP capturing an electron. This probability is size-dependent. Megner and Gumbel (2009) assume a probability of zero for sizes less than 0.25 nm, a probability of 1 for sizes larger than 1.5 nm and for sizes between 0.25 and 1.5 nm, they assume a probability that increases linearly. Baumann

et al. (2013) applies the electron attachment efficiency ($\gamma_{charging}$) from Megner and Gumbel (2009) to the ionospheric model. Megner and Gumbel 2009 proposed this charging efficiency based on a laboratory study on the charging of water ice clusters. The size dependence of the charging efficiency is probably a function of dust composition. Therefore, we add two alternative scenarios of this efficiency to study its possible impact on the electron heating. Table 2 summarizes the different electron attachment efficiencies applied in our study. The computation scheme includes chemical reactions like the standard plasma

reactions for electrons and ions, plasma capture reactions by MSP and photo reactions such as photo ionization and photo detachment of MSP. The standard plasma reactions include ionization, dissociative recombination, electron attachment to neutrals, electron detachment from negative ions and ion-ion recombination. Figure 3 shows a schematic of the underlying ionospheric model. By solving the time-dependent rate equations for the six species, the ionospheric model computes number densities of electrons, ions and MSPs. The rate equations describes how the concentration of a given species varies with time

by looking at the local production rate and local loss rate. The modelling is done with and without the MSPs, as a comparison. For the initial conditions, the following parameters are taken from the Sodankylä Ion Chemistry (SIC) model (Turunen et al.,

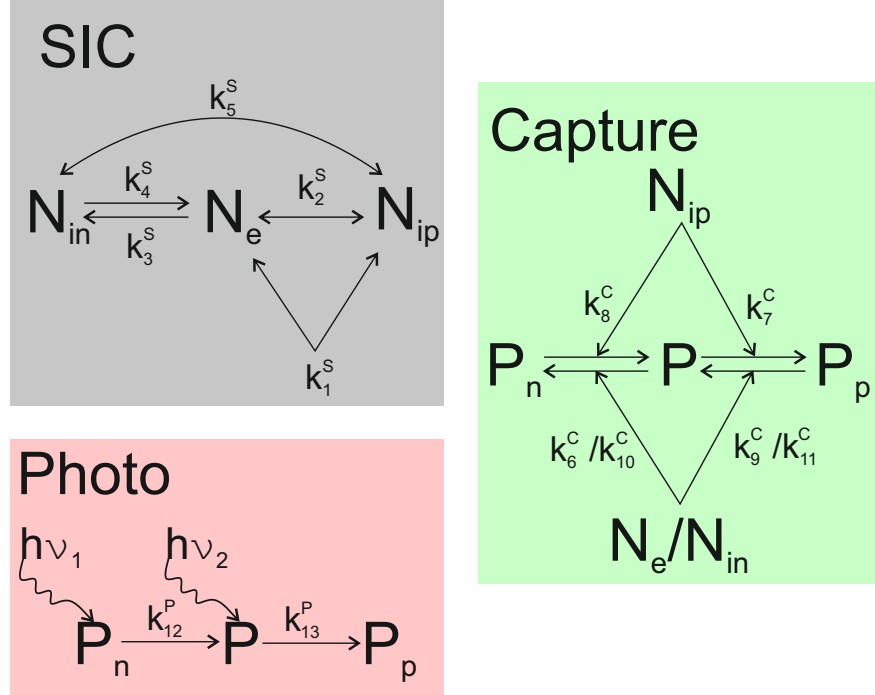

**Figure 3.** Schematic of the underlying ionospheric model. Grey shaded reactions are SIC reactions rates generalized to the reduced set of ionospheric constituents ($N_{in}$- negative ions, $N_e$- electrons, $N_{ip}$- positive ions), green shaded are charge carrier capture processes by MSP ($P_n$- negative MSP, $P$- neutral MSP, $P_p$- positive MSP) and red shaded are the photo detachment and photo ionization of MSP, where the wiggly arrow indicates the incoming solar photon that detaches an electron from the surface of a neutral or negatively charged MSP. The $k_1 - k_{13}$ are reaction rates coefficients. For details on the individual reactions see Baumann et al. (2013).

1996): number densities of electrons, positive ions and negative ions, the temperature of ions and electrons, the reaction rate coefficients for the standard plasma reactions and average ion mass. The SIC model was run for conditions on 8. September 2010 at Andenes, Norway, 69° North and 16° East at 23:55 LT (night conditions) and 12:15 LT (day conditions). We ran

the ionospheric model with different MSP number density profiles: Autumn conditions (8. September), winter conditions (1. January) and summer conditions (20. July). The model runs with different MSP number densities are performed with the following autumn ionospheric conditions: autumn MSP distribution for night and day conditions, winter MSP distribution for night conditions and summer MSP distribution for day conditions. The MSP winter and summer distribution come from Megner et al. (2008).

**Table 2.** The different electron attachment efficiencies ($\gamma_{charging}$), where $r$ is the MSP radius. 'MSP, I': the probability is 1 for all MSP sizes. 'MSP, II': the probability is zero for MSP sizes below 0.25 nm, between 0.25 to 1.5 nm the probability increases linearly and for sizes larger than 1.5 nm the probability is 1. 'MSP, III': the probability is zero for MSP size below 1.5 nm and 1 for sizes larger than 1.5 nm. Note that 'MSP, II' come from Megner and Gumbel (2009).

| MSP, I | MSP, II | MSP, III |
|---|---|---|
| $\gamma_{charging} = \begin{cases} 1, & \text{for all } r \end{cases}$ | $\gamma_{charging} = \begin{cases} 0, & \text{for } r < 0.25 \text{ nm} \\ 0.8 \cdot r - 0.2, & \text{for } 0.25 \le r \le 1.5 \text{ nm} \\ 1, & \text{for } r > 1.5 \text{ nm} \end{cases}$ | $\gamma_{charging} = \begin{cases} 0, & \text{for } r \le 1.5 \text{ nm} \\ 1, & \text{for } r > 1.5 \text{ nm} \end{cases}$ |

## 4 Results

### 4.1 Night conditions

This section presents results for the electron temperature modelled during artificial heating with and without the presence of MSPs for night condition. The main results are that from 80 km and above the heated electron temperature is higher when MSPs are present, and this applies to all cases with different frequencies and ERP. In Fig. 4 we show results for electron density influenced by MSPs. As a comparison, we ran the model without the influence of MSPs. We see in Fig. 4 that there is a reduction in electron density, an electron bite-out, due to the presence of MSPs, predominantly between heights 80-100 km. There is an electron bite-out between 70-80 km, but it is significantly smaller than between 80-100 km. Between 100-120 km, the electron bite-outs are not present. We see that electron bite-outs change the height profile of the electron density.

Figure 5 presents results for the heated electron temperature for cases 1-4. The heated electron temperature is computed with the electron density from Fig. 4. In Fig. 5 we see that the electron temperature is higher for altitudes above 80 km when MSPs are present. The shape of the height profile varies as well, where the heated electron temperature decreases more slowly when MSPs are present so that the shape of the height profile is more flat. Without MSPs, the electron temperature decreases faster and it has a different overall shape. Below 80 km, the heated electron temperature is the same with and without MSPs. A comparison of the four different cases shows similar results for the heated electron temperature in Fig. 5. The electron temperature is higher when MSPs are present for all five different cases and the shape is also similar. We also see that a higher ERP results in a higher electron temperature, where $T_e$ reaches almost 3000 K for case 4 with ERP 1200 MW.

Figure 6 shows the absolute difference between the heated electron temperature modelled with and without MSPs, i.e. how much higher the heated electron temperature is with MSPs compared to without MSPs. The difference in $T_e$ increases from 80 km and reaches a maximum between 90-100 km. From 100 km and on, the difference in $T_e$ starts to decrease. The difference in $T_e$ increases for higher ERP. For case 4 with a frequency of 7.5 MHZ and ERP 1200 MW, the difference in $T_e$ at around 100 km is almost 1000 K, while for case 3 with a frequency of 5.5 MHZ and ERP 600 MW the difference in $T_e$ at around 95 km is 700 K. For the lower ERP of 200 MW with frequency 4 MHZ (case 1) or ERP of 300 MW with frequencies 5.5 MHZ (case 2), the difference in $T_e$ is between 200-500 K at 95 km.

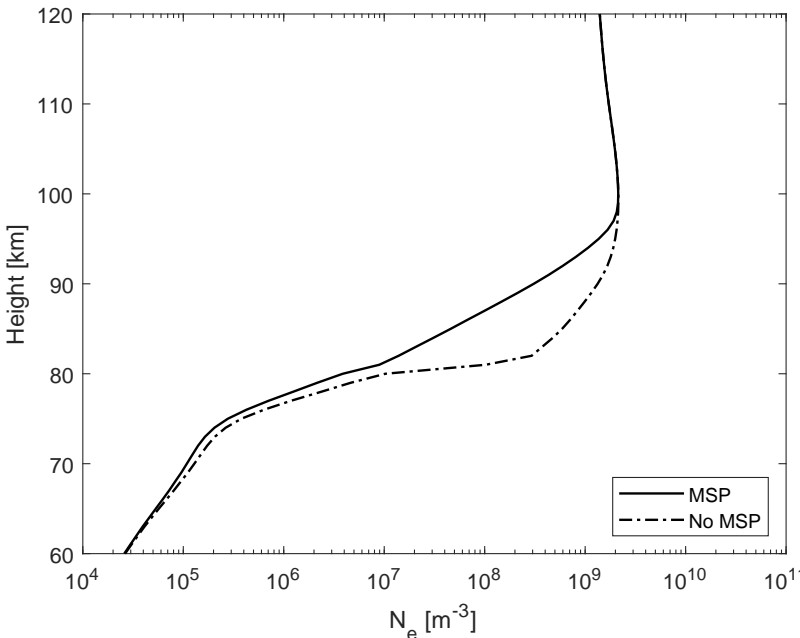

**Figure 4.** Electron density during night conditions, where the electron density comes from the ionospheric model. The legend shows model run with and without the MSPs.

In Fig. 5 a small feature appears in some of the plots when the electron temperature is around 1500 K. The location of the feature appears at different altitudes, varying between 80-110 km. Out of the four different cases that we considered for comparison (4 cases with MSPs and 4 cases without MSPs, so 8 all together), the feature appears in 4 out of 8 plots, 1 with MSPs and 3 without MSPs.

In Fig. 7 we show results for MSP winter distribution (night ionospheric conditions) and MSP summer distribution (day ionospheric conditions). Panel a) shows electron density, while panel b) shows heated electron temperature. For model run with MSPs, a comparison of electron densities in panel a) of Fig. 7 and Fig. 4 show a slightly higher electron depletion below 80 km in the winter case compared to the autumn case. However, above 80 km, the electron depletion is higher for the autumn case. For the winter case, the reduction in electron density extends to around 90 km, while it extends to around 100 km for the autumn case. In Fig. 5, the heated electron temperature for the autumn case is higher above 80 km compared to the winter case in panel b) of Fig. 7; the difference is less than 280 K. Our results in Fig. 7 for the summer case are quite similar to the autumn case. This applies to the behaviour of both the electron density and the heated electron temperature. The difference between the heated electron temperature for the summer case and the autumn case is less than 8 K.

Figure 8 shows model results for different cases of electron attachment efficiencies of MSPs, where panel a) shows electron density and panel b) shows heated electron temperature. In this study, we concentrate on three different scenarios for size-dependent probabilities of electron attachment to MSPs: 'MSP, I' - the probability is 1 for all MSP sizes. 'MSP, II' - the

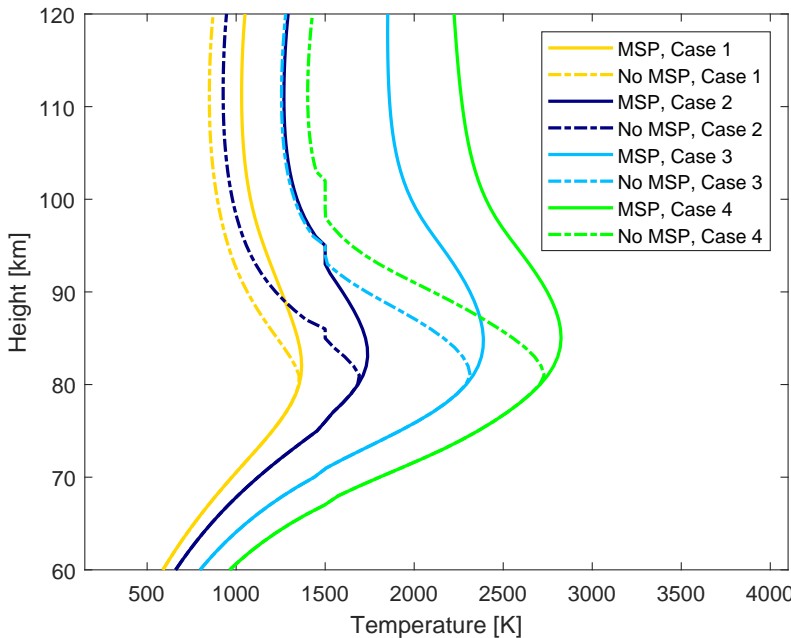

**Figure 5.** Night condition results for modelled electron temperature during heating as a function of height for cases 1-4. The legend shows model run with and without the MSPs and for the different cases 1-4.

probability is zero for MSP sizes below 0.25 nm, between 0.25 nm to 1.5 nm the probability increases linearly and for sizes larger than 1.5 nm the probability is 1. 'MSP, III' - the probability is zero for MSP size below 1.5 nm and 1 for sizes larger than 1.5 nm. See also table 2 for more details. We see in panel a) that the magnitude of the reduced electron density depends on the electron attachment efficiency. In the case 'MSP I', the electron density is severely reduced because more MSPs are available to be charged. If there is no charging for sizes below 1.5 nm, the electron density is quite similar to the electron density when no MSPs are present. This applies to the electron temperature in panel b) as well. For the case where the probability is 1 for all sizes (MSP, I), the heated electron temperature remains almost constant from 90-120 km. The temperature difference between 'MSP, I' and 'MSP, III' goes up to 750 K.

## 4.2 Day conditions

This section presents results for electron temperature modelled during artificial heating with and without the presence of MSPs for day conditions. Panel a) of Fig. 9 shows electron density with and without MSPs. As for night conditions the electron density comes from the ionospheric model. We see that the electron bite-outs are much smaller in Fig. 9 compared to the night condition results in Fig. 4. Panel b) of Fig. 9 shows the heated electron temperature for cases 1-4. We see that the heated electron temperature is the same with and without MSPs. We find that the absolute difference between the electron temperature modelled with and without MSPs is less than 25 K for all cases 1-4. Compared to night conditions, the day conditions electron

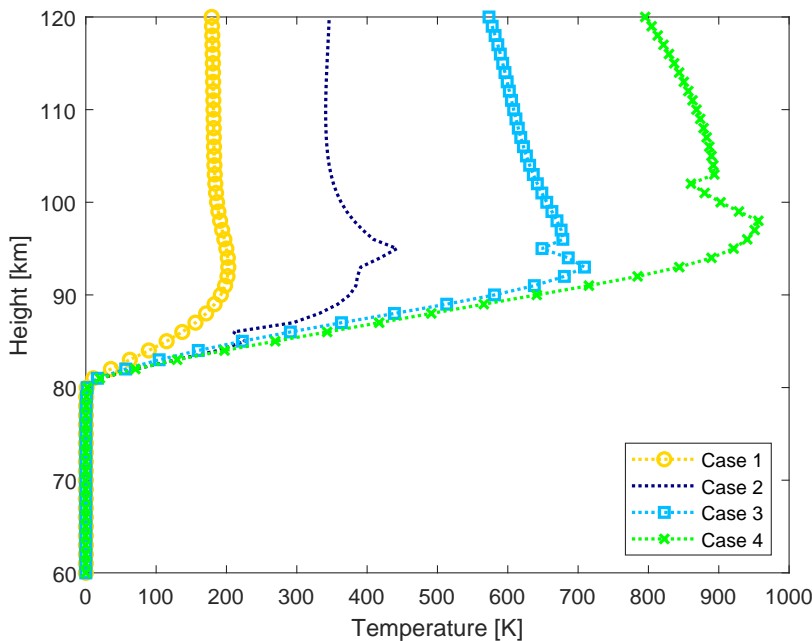

**Figure 6.** Absolute difference between the electron temperature modelled with and without the MSPs during night conditions. The legend shows model run for the different cases 1-4.

temperature is lower. For cases 1-3 during day conditions, the electron temperature is below 2000 K for all altitudes. At around 90 km, the electron temperature is back to the neutral temperature for all cases 1-4.

## 5  Discussion

Both Kero et al. (2007) and Senior et al. (2010) found that current models most likely overestimate artificial heating in the D-region compared to observations. Why the models overestimate artificial heating in the D-region remains an open question.

Kero et al. (2007) studied how artificial heating influences cosmic radio noise absorption. Their study showed that the observed enhancement of cosmic radio noise absorption during heating is lower than predicted theoretically. Senior et al. (2010) used a cross-modulation technique with the EISCAT radar. They found that the model overestimates the diagnostic wave absorption. An explanation for the discrepancy between models and observations suggested by Senior et al. 2011 is that the modeled heater ERP is lower than predicted because the assumption of a perfect reflecting ground around the antenna might not be applicable.

Senior et al. 2011 found that the overestimation is reduced when modelling the ERP with more realistic ground assumptions.

In the study by Senior et al. (2010), the authors note that electron bite-outs located at PMSE layer altitudes might influence the model, but that the influence is probably small, because the bite-outs are located too high in altitude. They investigate the influence of the electron bite-outs by scaling the whole electron density profile with a factor of 2 or 0.5. However, they do not

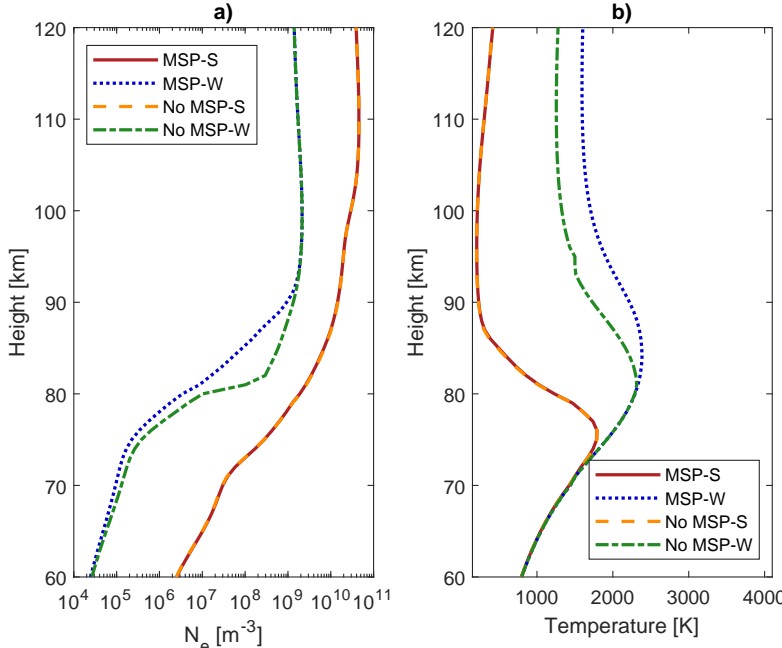

**Figure 7.** Results for MSP winter distribution (night ionospheric conditions) and MSP summer distribution (day ionospheric conditions). The frequency is 5.5 MHz and the ERP is 600 MW. Panel a) shows electron density and panel b) shows heated electron temperature. We also show model run without MSPs. In the legend, 'S' stands for summer conditions, while 'W' stands for winter conditions.

include the height variation of the electron density profile when electron bite-outs are present. We find that electron bite-outs are only present at certain altitudes. The magnitude of the electron bite-outs varies within these altitudes, for instance, the electron bite-out is significantly larger between 80-100 km compared to between 70-80 km. In our study, we have modelled the electron temperature during heating and included the height variation of the electron bite-outs. We have included the height variation of electron bite-outs by using the ionospheric model with MSPs, which presents a simplified model of the D-region by including height and size-dependent MSP distribution in a reaction scheme with electrons, ions and neutral and charged MSPs. This enables us to have a more realistic representation of the height variation of the electron bite-outs. In a future study we will make a detailed comparison of our results to observations of the electron temperature during heating. This detailed comparison can investigate if the presence of MSPs explain the discrepancy between model and observations.

Figure 5 for night conditions show that the electron temperature is higher and decreases more slowly when MSPs are present. An explanation for why the heated electron temperature decreases more slowly is that with electron bite-outs at certain altitudes, the heating above these heights will be increased since less of the wave energy is absorbed within the electron bite-outs. The absorption of wave energy depends on electron density and the absorption decreases with decreasing electron density. We see this effect in Fig. 10 , which shows absorbed radio wave energy as a function of height. Here, less wave energy is absorbed when MSPs are present. More wave energy is absorbed at higher altitudes, slightly above where the electron bite-outs are largest in

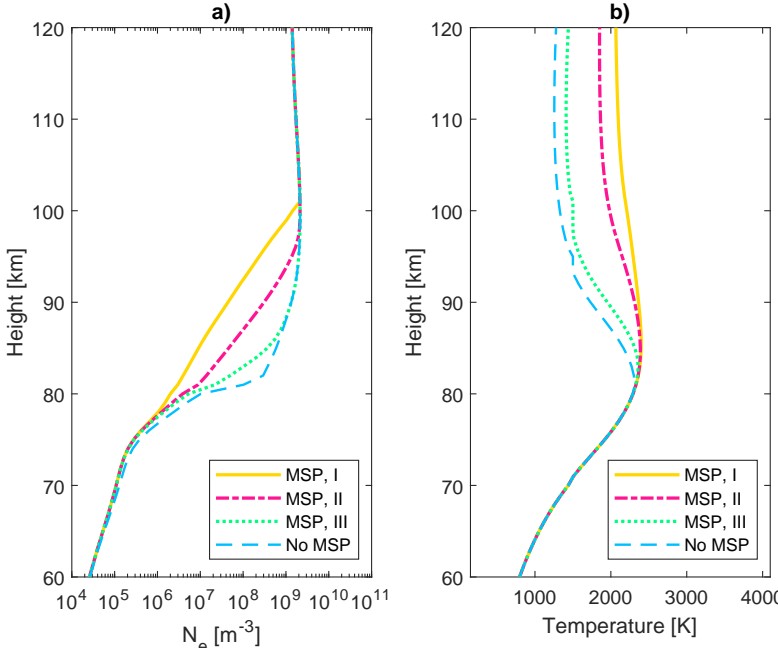

**Figure 8.** Result for different electron attachment efficiencies. Panel shows a) electron density and panel b) shows heated electron temperature. The legend describes the different size-dependent probabilities for electron attachment to MSP: 'MSP, I' - the probability is 1 for all MSP sizes. 'MSP, II' - the probability is zero for MSP sizes below 0.25 nm, between 0.25 nm to 1.5 nm the probability increases linearly and for sizes larger than 1.5 nm the probability is 1. 'MSP, III' - the probability is zero for MSP size below 1.5 nm and 1 for sizes larger than 1.5 nm. See table 2 for more details on the different electron attachment efficiencies. We also show model run without MSP.

magnitude. The cooling rates also depend on electron density and decrease at higher altitudes due to a lower electron-neutral collision frequency since the neutral density is lower. The electron cooling - heating equality is reached at higher electron temperatures as more wave energy remains in the MSPs case compared to the case without MSPs. An electron bite-out at lower altitudes can lead to an increased electron temperature at higher altitudes above.

Our results in Fig. 8 for the different electron attachment efficiencies indicates that the heated electron temperature height profile is very dependent on the amount of chargeable MSPs. Increasing the amount of chargeable MSPs leads to a nearly vanishing electron density at altitudes between 80 and 100 km. This aspect of MSPs is not very well known and could be investigated further. Note that the electron density profile in case 'MSP,I' might be unrealistically low since it is around one order of magnitude below the electron density measured during ECOMA-7 rocket flight (between 80-95 km), which is the lowest electron density ever measured at auroral latitudes (Friedrich et al., 2012). Given that the case 'MSP,I' is indeed very unlikely, indicates that there are either not that many small MSPs (sizes below 0.25 nm) or that the smaller MSPs are not charged. The modelling with different electron attachment efficiencies (different charging) and with different MSP number

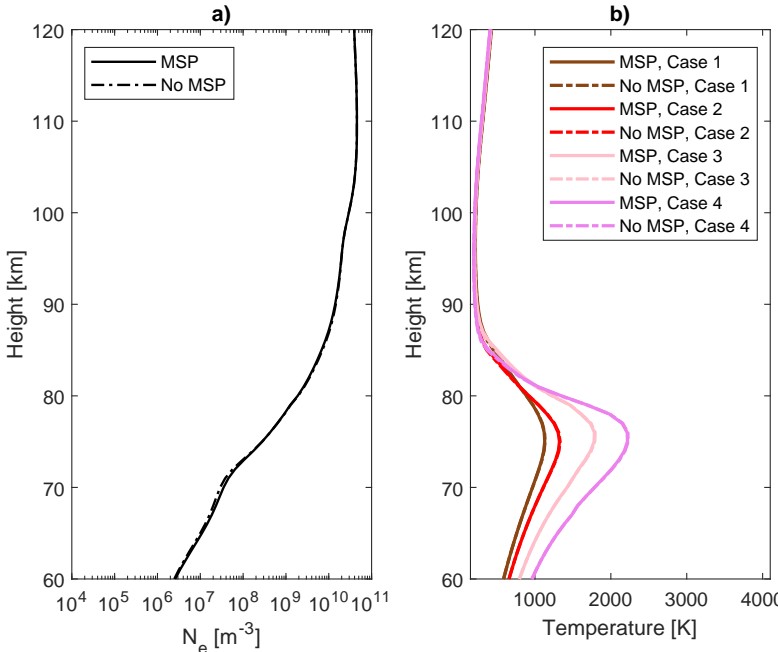

**Figure 9.** Results for day conditions. Panel a) shows the electron density, which come from the ionospheric model. Panel b) shows the the modelled electron temperature during heating as a function of height. The legend show model run with and without the MSP, as well as model run for the different cases 1-4.

density profiles indicates that the night-time D-region electron temperature varies with the number of chargeable MSP, which again varies with the MSP number density and the charging efficiency.

The results of our study show that the frequency of the transmitted radio wave only plays a minor role, lower frequency only slightly shifts the start of the heated ionosphere to a higher altitude. We also see that the increased electron temperature due to the presence of MSPs extends up to 120 km in the E-region. Our model for the heated electron temperature might not be applicable for the E-region, however, this is beyond the scope of this paper. The results from this study agree with Kassa et al. (2005), where an electron bite-out inserted as a linearly decreasing 'toy model' between 84-86 km during PMSE conditions resulted in an increased modelled electron temperature within and above the electron bite-out.

Panel b) of Fig. 9 shows that day condition electron temperature is the same with and without MSPs. This indicates that for day condition, MSPs are less important for the heated electron temperature. A higher ionization level, and thus a much higher electron density, means that loss of electrons, like electron attachment to MSPs, is less important. Generally, the electron temperature is lower for day conditions. This is because the electron density is higher during the day, also at lower heights. The electron density of panel a) in Fig. 9 is $2.5 \cdot 10^6$ m$^{-3}$ at 60 km for day conditions, while for night conditions the electron density in Fig. 4 is $2.6 \cdot 10^4$ m$^{-3}$ at 60 km. With a higher electron density as during day conditions, the radio wave energy is absorbed already at lower heights.

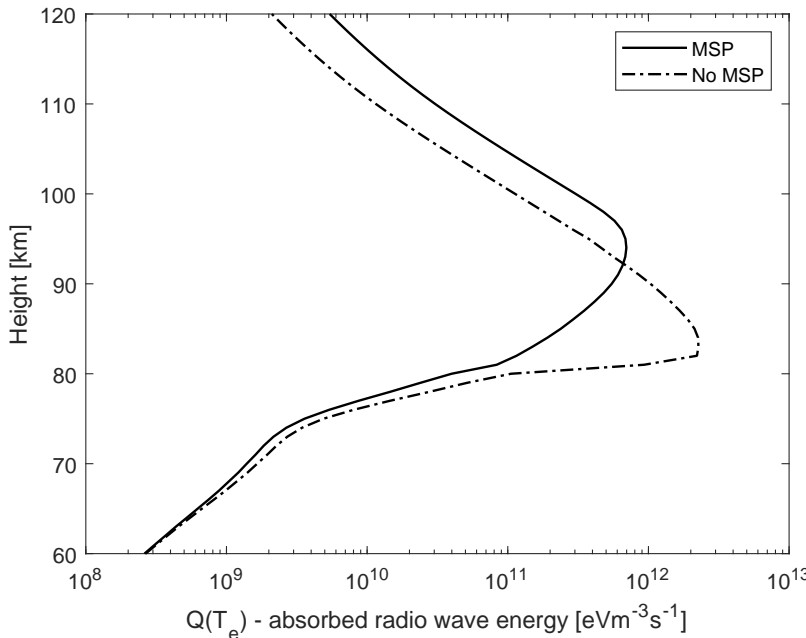

**Figure 10.** Absorbed radio wave energy $Q (= L)$ as a function of height for night condition and case 2 (5.5 MHz and ERP 600 MW). The legend show model run with and without the MSP. We show this figure to illustrate how the absorbed power varies with and without MSPs.

In Fig. 5 there is a feature in some of the plots of the heated night condition electron temperature. This feature can resemble a small second maximum, or it might just be an artefact. It appeared when we included the temperature dependence of the cooling rates for vibrational excitation of molecular nitrogen; the values from Pavlov (1998a) that we use are different for the temperatures $300 \leq T_e \leq 1500$ K and for those $T_e > 1500$ K. The feature that we note is at electron temperature around 1500

315 K. The feature disappears if we apply the same values for vibrational excitation of molecular nitrogen over the entire range of temperatures and disregard the difference for the $T_e \leq 1500$ K case. Kero et al. (2008) found a second maximimum in the EISCAT incoherent scatter observations for the heated electron temperature in the D-region, which they could not explain. The feature in Fig. 5 might be a second maximum or it might be an artefact caused by problems in the numerical modelling when switching between values for $T_e \leq 1500$ K and $T_e > 1500$ K. Whether the feature is an artefact or not is unknown at the

320 present and can be investigated further. The feature is not seen in the day condition electron temperature of panel b) in Fig. 9.

## 6   Conclusions

The presented model calculations show that the presence of MSPs can influence the electron temperature during artificial heating. The influence of the MSPs varies with ionospheric conditions. For night conditions, the results show a higher heated electron temperature above altitudes of 80 km when MSPs are present. We found differences of up to 1000 K in temperature

for calculations with and without MSPs. Below 80 km of altitude for night conditions the difference in temperature are small for model calculations with and without MSPs. For day conditions, the difference between the heated electron temperature with MSPs and without MSPs is less than 25 K. This study indicates that MSPs can influence both the magnitude and shape of the heated electron temperature above 80 km, however this depends on ionospheric conditions.

Furthermore, we model with different MSP number density profiles for autumn, summer and winter. The results show 280 K hotter night-time electron temperature for autumn compared to winter, while for the daytime electron temperature, the autumn case is 8 K cooler than the summer case. However, this varies with altitude. Finally, our results shows that the electron attachment efficiency influences the heated electron temperature by impacting the amount of chargeable MSPs. In future studies, we will model the D-region electron temperature during artificial heating with a non-Maxwellian electron velocity distribution, possibly combining it with our study about artificial heating and MSPs.

*Code and data availability.* A function that computes the electron temperature and radio wave intensity during artificial heating, including the electron cooling rates, is available at Myrvang (2021). Data sets, which contain electron density altitude profiles from the background ionospheric model (Baumann et al., 2013), can be found at Baumann and Myrvang (2021).

### Appendix A: MSP number density profiles

Figure A1 shows the different MSP number density profiles: panel a) shows the MSP autumn case (Megner et al., 2006) for 8. September, panel b) shows the MSP winter case (Megner et al., 2008) for 1. January and panel c) shows the MSP summer case (Megner et al., 2008) for 20. July. The MSP number density profile for autumn and winter is quite similar. However, the difference between the winter and summer case is quite significant, particularly for the larger sizes above 5 nm, which is more abundant for the summer case and extends to a higher altitude as well.

### Appendix B: Electron cooling rates

The electrons lose energy through collisions with the neutral gases. The dominant cooling processes related to $[N_2]$ and $[O_2]$ are the energy transfer via vibrational and rotational excitation [cf. Rietveld et al. (1986); Gustavsson et al. (2010)]. Even though the concentration of atomic oxygen is very small between 60-100 km (as discussed by Senior et al. 2010), we will include electron cooling rates for atomic oxygen [O] through the impact excitation of fine structure levels of its ground state (see Pavlov and Berringston (1999) and references given there). We do this because our modelling is between 60-120 km, and the concentration of atomic oxygen increases above 100 km. At 120 km, the concentration of atomic oxygen is in the same order of magnitude as the concentration of molecular oxygen. We here repeat the cooling rates that are used. The sum of the electron cooling rates are the energy loss function, given as:

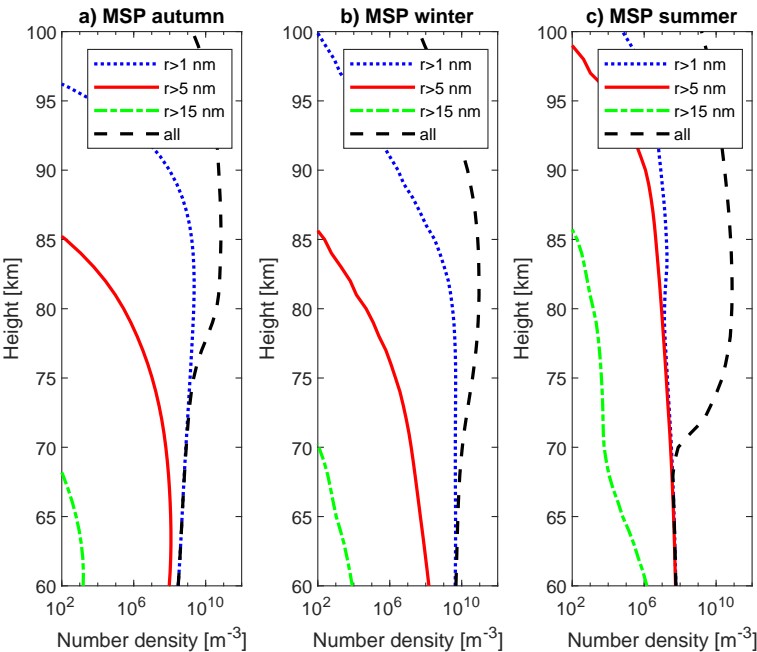

**Figure A1.** Different MSP number density profiles for a) the autumn case, b) the winter case and c) the summer case. The autumn case is from Megner et al. (2006), while the winter and summer case is from (Megner et al., 2008).

$$L(T_e) = L_{fs}(O) + L_{vib}(N_2) + L_{rot}(N_2) + L_{vib}(O_2) + L_{rot}(O_2) + L_{el}(N_2) + L_{el}(O_2) + L_{el}(O) \tag{B1}$$

The unit of $L(T_e)$ are in $Jm^{-3}s^{-1}$.

To describe the excitation of fine structure levels of atomic oxygen, we use Eq. 15 from Pavlov and Berringston (1999):

$$
\begin{aligned}
L_{fs}(O) = N_e[O]D^{-1}(S_{10}\{1 - \exp[98.9(T_e^{-1} - T_n^{-1})]\} \\
+ S_{20}\{1 - \exp[326.6(T_e^{-1} - T_n^{-1})]\} \\
+ S_{21}\{1 - \exp[227.7(T_e^{-1} - T_n^{-1})]\})
\end{aligned}
\tag{B2}
$$

The units of equation B2 is $eVcm^{-3}s^{-1}$ and $T_n$ is the neutral temperature. Both $T_e$ and $T_n$ are in K. The equation is based on assuming that the electron velocity distribution is Maxwellian. The terms $D$, $S_{21}$, $S_{20}$ and $S_{10}$ are:

$$D = 5 + \exp(-326.6 \cdot T_n^{-1}) + 3\exp(-227.7 \cdot T_n^{-1}) \tag{B3}$$

$$S_{21} = 1.863 \cdot 10^{-11} \tag{B4}$$

$$S_{20} = 1.191 \cdot 10^{-11} \tag{B5}$$

$$S_{10} = 8.249 \cdot 10^{-16} \cdot T_e^{0.6} \exp(-227.7 \cdot T_n^{-1}) \tag{B6}$$

The $S_{ij}$ denote the transitions between the three fine structure levels of the atomic oxygen ground state.

For vibrational excitation of molecular nitrogen, we use Eq. 11 from Pavlov (1998a) for a Boltzmann distribution:

$$
\begin{aligned}
L_{vib}(N_2) = & N_e[N_2]\{1 - \exp(-E_1/T_{vib})\} \\
& \times \sum_{v=1}^{10} Q_{0v}\{1 - \exp[vE_1(T_e^{-1} - T_{vib}^{-1})]\} \\
+ & N_e[N_2]\{1 - \exp(-E_1/T_{vib})\}(\exp(-E_1/T_{vib})) \\
& \times \sum_{v=2}^{9} Q_{1v}\{1 - \exp[(v-1)E_1(T_e^{-1} - T_{vib}^{-1})]\}
\end{aligned}
\tag{B7}
$$

where $E_1 = 3353$ K is the energy of first vibrational level of $[N_2]$ and we assume that the vibrational temperature is equal to the neutral temperature. The units of $L_{vib}(N_2)$ is eVcm$^{-3}$s$^{-1}$. Here, $Q_{0v}$ describes excitation transitions from ground states and $Q_{1v}$ describes excitation transitions from the first vibrational state. For $Q_{0v}$ and $Q_{1v}$, we implement Eq. 19 and Eq. 20 from Pavlov (1998a), respectively:

$$\log Q_{0v} = A_{0v} + B_{0v}T_e + C_{0v}Te^2 + D_{0v}Te^3 + F_{0v}Te^4 - 16 \tag{B8}$$

$$\log Q_{1v} = A_{1v} + B_{1v}T_e + C_{1v}Te^2 + D_{1v}Te^3 + F_{1v}Te^4 - 16 \tag{B9}$$

where the coefficients $A_{0v}, B_{0v}, C_{0v}, D_{0v}, F_{0v}$ to compute $Q_{0v}$ and $A_{1v}, B_{1v}, C_{1v}, D_{1v}, F_{1v}$ to compute $Q_{1v}$ come from tables in Pavlov (1998a). For $Q_{0v}$ from Table 1 for $300 \leq T_e \leq 1500$ K and from Table 2 for $T_e > 1500$ K. For $Q_{1v}$ from Table 3 for $1500 \leq T_e \leq 6000$ K. However, there is no table for $Q_{1v}$ for $T_e < 1500$ K. Both $Q_{0v}$ and $Q_{1v}$ have units eVcm$^3$s$^{-1}$. Rotational excitation of molecular nitrogen come from Eq. A2 in Pavlov (1998a):

$$C = 3.51 \cdot 10^{-14} \tag{B10}$$

$$L_{rot}(N_2) = C[N_2]N_e(T_e - T_n)Te^{-0.5} \tag{B11}$$

The units of $C$ and $L_{rot}(N_2)$ are $\mathrm{eVcm^3s^{-1}K^{-0.5}}$ and $\mathrm{eVcm^{-3}s^{-1}}$, respectively.

For vibrational excitation of molecular oxygen we use Eq. 8 from Pavlov (1998b), which assumes a Boltzmann distribution:

$$L_{vib}(O_2) = N_e[O_2]\sum_{v=2}^{7} Q_{0v}^*\{1 - \exp[vE_1(T_e^{-1} - T_{vib}^{-1})]\} \tag{B12}$$

in units $\mathrm{eVcm^{-3}s^{-1}}$ and where $E_1 = 2239$ K is the energy of the first vibrational level of $[O_2]$ and we set $T_{vib} = T_n$. Here $Q_{0v}$ describes excitation transitions from ground states. $Q_{0v}$ come from Eq. 11 in Pavlov (1998b):

$$Q_{0v}^* = A_v \exp\{(1 - B_v T e^{-1})(C_v + D_v \sin[F_v(T_e - G_v)])\} \tag{B13}$$

where the coefficients $A_v, B_v, C_v, D_v, F_v, G_v$ as a function of vibrational level come from Table 1 of Pavlov (1998b). For rotational excitation of $[O_2]$ we use Eq. 16, also from Pavlov (1998b):

$$C_{O_2} = 5.2 \cdot 10^{-15} \tag{B14}$$

$$L_{rot}(O_2) = C_{O_2}[O_2]N_e(T_e - T_n)T_e^{-0.5} \tag{B15}$$

where $C_{O_2}$ have units $\mathrm{eVcm^3s^{-1}K^{-0.5}}$ and $L_{rot}(O_2)$ has units $\mathrm{eVcm^{-3}s^{-1}}$. For elastic collisions between electrons and neutrals (molecular nitrogen, molecular oxygen and atomic oxygen, respectively) we implement Eq. 43a, 43b, 43c from Schunk and Nagy (1978):

$$L_{el}(N_2) = N_e[N_2]1.77 \cdot 10^{-19}T_e(T_e - T_n)(1 - 1.21 \cdot 10^{-4}T_e) \tag{B16}$$

$$L_{el}(O_2) = N_e[O_2]1.21 \cdot 10^{-18}\sqrt{T}_e(T_e - T_n)(1 + 3.6 \cdot 10^{-2}\sqrt{T}_e) \tag{B17}$$

$$L_{el}(O) = N_e[O]7.9 \cdot 10^{-19}\sqrt{T}_e(T_e - T_n)(1 + 5.7 \cdot 10^{-4}T_e) \tag{B18}$$

In Fig. B1 and Fig. B2 we present height profiles for electron cooling rates for night conditions. We show electron cooling rates for a heated electron temperature. Figure B1 shows cooling rates where MSP are present, while Fig. B2 shows cooling rates where MSP are not present. The frequency is 5.5 MHz and ERP is 600 MW.

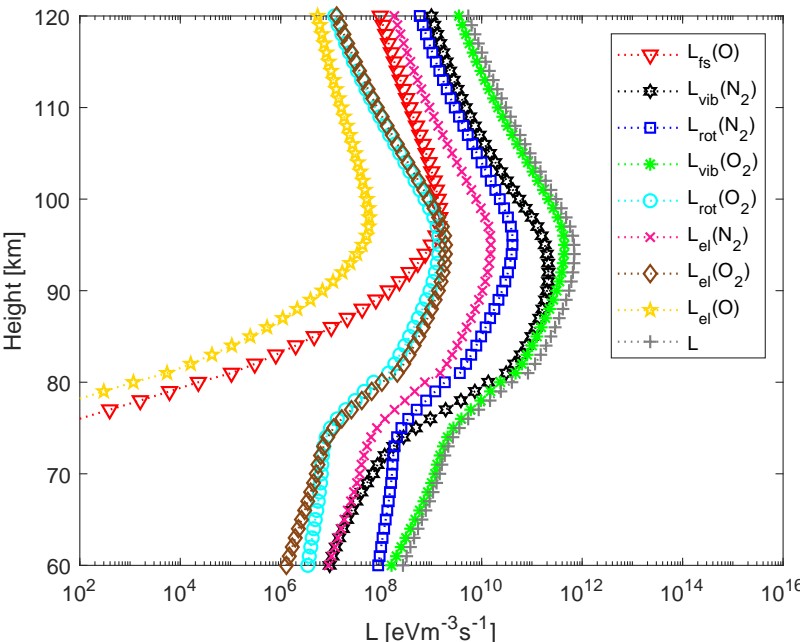

**Figure B1.** Night condition electron cooling rates with MSP as a function of height for a heated electron temperature. The frequency is 5.5 MHz and ERP is 600 MW. The legend shows the different cooling rates as described in this section.

*Author contributions.* Margaretha Myrvang made the artificial heating program and prepared the initial manuscript. Carsten Baumann developed the ionospheric model. Ingrid Mann suggested the topic and supervised the project. All authors contributed to the preparation of the manuscript.

*Competing interests.* Ingrid Mann is editor-in-chief and topical editor of ANGEO.

*Acknowledgements.* We would like to thank Ove Havnes for providing us with a program from Meseret Kassa, which computes the electron temperature during heating. We have compared our results to the program from Meseret Kassa. In addition, we would like to thank Antti Kero for suggesting to look at how the presence of MSP can influence artificial heating. Lastly, we would like to thank Björn Gustavsson for helping us with the theory of artificial heating.

This work was supported by the Research Council of Norway through grant numbers: The Mesospheric Dust in Small Size Limit NFR 275503. The publication charges for this article have been funded by a grant from the publication fund of UiT The Arctic University of Norway.

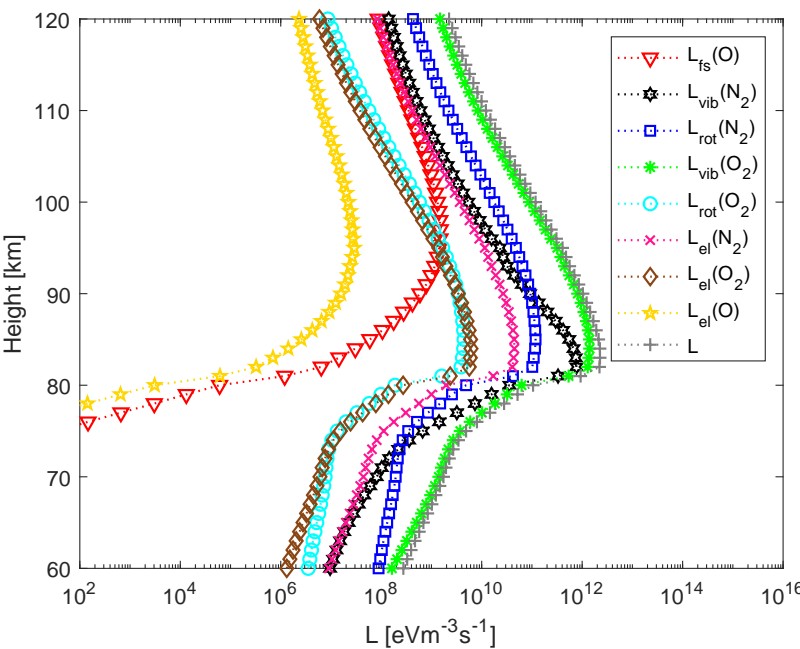

**Figure B2.** Night condition electron cooling rates without MSP as a function of height for a heated electron temperature. The frequency is 5.5 MHz and ERP is 600 MW. The legend shows the different cooling rates as described in this section.

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
