# Peer review of "Modelling the influence of meteoric smoke particles on artificial heating in the D-region"

_Annales Geophysicae, 2021_

## Referee Comment (RC2)

[referee-annotated manuscript omitted]

---

## Author Response (AR1)

Response to Anonymous Referee #1

We would like to thank the Anonymous Referee #1 for their constructive comments and suggestions.

As suggested by the referee, we include a study on how the more uncertain aspects of meteoric smoke particles influence HF heating of electrons. In the revised version, we discuss the results for different values of the electron attachment efficiency and for summer and winter meteoric smoke conditions. We have included all the referee's suggestions for the technical comments. We address all the technical comments below. The referee report is marked in bold, while our answers are marked in normal text.
* * *
We have added a paragraph to line 18 (abstract) to summarize our findings on the different values of the electron attachment efficiency and for summer and winter meteoric smoke conditions:
"We also investigate model runs using MSP number density profiles for autumn, summer and winter. The night-time electron temperature is expected to be 280 K hotter in autumn than during winter conditions, while the sunlit D-region is 8 K cooler for autumn MSP conditions than for the summer case, depending on altitude. Finally, an investigation of the electron attachment efficiency to MSPs shows a significant impact on the amount of chargeable dust and consequently on the electron temperature."

In line 50 (section 1, introduction) we have added:
"In addition, we investigate the seasonal variation of the MSPs abundance, as well as the role of the electron attachment efficiency to MSPs for the heated electron temperature."

We have added a paragraph describing the electron attachment efficiency to line 178 (section 3, background ionospheric model):
"For the charging of a MSP by electrons, the electron attachment efficiency is the probability of a MSP capturing an electron. This probability is size-dependent. Megner and Gumbel 2009 assume a probability of zero for sizes less than 0.25 nm, a probability of 1 for sizes larger than 1.5 nm and for sizes between 0.25 and 1.5 nm, they assume a probability that increases linearly. Baumann et al. 2013 applies the electron attachment efficiency ($\Upsilon_{charging}$) from Megner and Gumbel 2009 to the ionospheric model. Megner and Gumbel 2009 proposed this charging efficiency based on a laboratory study on the charging of water ice clusters. The size dependence of the charging efficiency is probably a function of dust composition. Therefore, we add two alternative scenarios of this efficiency to study its possible impact on the electron heating. Table 2 summarizes the different electron attachment efficiencies applied in our study." Note that we have also added a table that describes the different electron attachment efficiencies.

At line 196 (section 3, background ionospheric model) we added a line that mentions the different MSP number density profiles (autumn/summer/winter conditions):
"We ran the ionospheric model with different MSP number density profiles: Autumn conditions (8. September), winter conditions (1. January) and summer conditions (20. July). The model runs with different MSP number densities are performed with the following autumn ionospheric conditions: autumn MSP distribution for night and day conditions, winter MSP distribution for night conditions and summer MSP distribution for day conditions. The MSP winter and summer distribution come from Megner et al. 2008."

We have added a paragraph to line 229 (section 4, results) that describes the results for MSP summer/winter conditions:

"In Fig. 7 we show results for MSP winter distribution (night ionospheric conditions) and MSP summer distribution (day ionospheric conditions). Panel a) shows electron density, while panel b) shows heated electron temperature. For model run with MSP, a comparison of electron densities in panel a) of Fig. 7 and Fig. 4 show a slightly higher electron depletion below 80 km in the winter case compared to the autumn case. However, above 80 km, the electron depletion is higher for the autumn case. For the winter case, the reduction in electron density extends to around 90 km, while it extends to around 100 km for the autumn case. In Fig. 5, the heated electron temperature for the autumn case is higher above 80 km compared to the winter case in panel b) of Fig. 7; the difference is less than 280 K. Our results in Fig. 7 for the summer case are quite similar to the autumn case. This applies to the behaviour of both the electron density and the heated electron temperature. The difference between the heated electron temperature for the summer case and the autumn case is less than 8 K. "

At line 239 (section 4, results) we have added a paragraph that describes the result for different electron attachment efficiencies:
"Figure 8 shows model results for different cases of electron attachment efficiencies of MSPs, where panel shows a) electron density and panel b) shows heated electron temperature. In this study, we concentrate on three different scenarios for size-dependent probabilities of electron attachment to MSP: 'MSP, I' - the probability is 1 for all MSP sizes. 'MSP, II' - the probability is zero for MSP sizes below 0.25 nm, between 0.25 nm to 1.5 nm the probability increases linearly and for sizes larger than 1.5 nm the probability is 1. 'MSP, III' -  the probability is zero for MSP size below 1.5 nm and 1 for sizes larger than 1.5 nm. See also table 2 for more details. We see in panel a) that the magnitude of the reduced electron density depends on the electron attachment efficiency. In the case 'MSP I', the electron density is severely reduced because more MSPs are available to be charged. If there is no charging for sizes below 1.5 nm, the electron density is quite similar to the electron density when no MSP are present. This applies to the electron temperature in panel b) as well. For the case where the probability is 1 for all sizes (MSP, I), the heated electron temperature remains almost constant from 90-120 km. The temperature difference between 'MSP, I' and 'MSP, III' goes up to 750 K."

At line 292 (section 5, discussion) we added the following:
"Our results in Fig. 8 for the different electron attachment efficiencies indicates that the heated electron temperature height profile is very dependent on the amount of chargeable MSPs. Increasing the amount of chargeable MSPs leads to a nearly vanishing electron density at altitudes between 80 and 100 km. This aspect of MSPs is not very well known and could be investigated further. Note that the electron density profile in case 'MSP,I' might be unrealistically low since it is around one order of magnitude below the electron density measured during ECOMA-7 rocket flight (between 80-95 km), which is the lowest electron density ever measured at auroral latitudes (Friedrich et al. 2012 ). Given that the case 'MSP,I' is indeed very unlikely, indicates that there are either not that many small MSP (sizes below 0.25 nm) or that the smaller MSPs are not charged. The modelling with different electron attachment efficiencies (different charging) and with different MSP number density profiles indicates that the night-time D-region electron temperature varies with the number of chargeable MSP, which again varies with the MSP number density and the charging efficiency."

At line 333 (section 6, conclusion):
"Furthermore, we model with different MSP number density profiles for autumn, summer and winter. The results show 280 K hotter night-time electron temperature for autumn compared to winter, while for the daytime electron temperature, the autumn case is 8 K cooler than the summer case. However, this varies with altitude. Finally, our results shows that the electron attachment efficiency influences the heated electron temperature by impacting the amount of chargeable MSPs.

In future studies, we will model the D-region electron temperature during artificial heating with a non-Maxwellian electron velocity distribution, possibly combining it with our study about artificial heating and MSPs."

\-\-\-\-\-\-\-\-\-\-\-\-\-\-\-\-\-\-\-\-\-\-\-\-\-\-

**Title: I suggest to remove the first "of".**
As suggested by the referee, we have removed the first "of" in the title since this improves the flow of the title. The title is changed to: "Modelling the influence of meteoric smoke particles on artificial heating in the D-region".

**Line 175: Reference to SIC model is lacking, and the acronym is never written out.**
Here the referee is right and we have added a reference to the SIC model and written out the acronym. We have changed the sentence to: "For the initial conditions, the following parameters are taken from the Sodankylä Ion Chemistry (SIC) model (Turunen et al., 1996)."

**Line 211: Well, technically there is a small bite-out, so I suggest to soften the wording.**
We agree with the referee, since there is a small electron bite-out at around 70 km. The sentence "We see that the electron bite-outs are not present in Fig. 7" is changed to: "We see that the electron bite-outs are much smaller in Fig. 9 compared to the night condition results in Fig. 4."

**Line 246: Remove "s" in "shows".**
We have now removed the s in "shows".
\-\-\-\-\-\-\-\-\-\-\-\-\-\-\-\-\-\-\-\-\-\-\-\-\-\-

In addition, we have made a number of modifications suggested by the other referee.

[Figure]

*Figure 7: Results for MSP winter distribution (night ionospheric conditions) and MSP summer distribution (day ionospheric conditions). The frequency is 5.5 MHz and the ERP is 600 MW. Panel a) shows electron density and panel b) shows heated electron temperature. We also show model run without MSP. In the legend, 'S' stands for summer conditions, while 'W' stands for winter conditions.*

[Figure]

*Figure 8: The different electron attachment efficiencies ($\Upsilon_{charging}$), where r is the MSP radius. 'MSP, I': the probability is 1 for all MSP sizes. 'MSP, II': the probability is zero for MSP sizes below 0.25 nm, between 0.25 to 1.5 nm the probability increases linearly and for sizes larger than 1.5 nm the probability is 1. 'MSP, III': the probability is zero for MSP size below 1.5 nm and 1 for sizes larger than 1.5 nm. Note that 'MSP, II' come from Megner and Gumbel 2009.*

We would like to thank the Anonymous Referee #2 for their thoroughly review of our manuscript and for the correction of spelling errors and grammar faults.

We have taken all the suggestions and comments from the referee into account. We address all the suggestions and comments below. The referee report is marked in bold, while our answers are marked in normal text.

**Line 117: The heating time is much less than 100 ms, it is less than 1 ms below 90 km (Stubbe et al., 1982).**
The referee is right. The heating time given by Stubbe et al., 1982 is 1 ms at 90 km and approximately 10 μs at 60 km. We will change the phrasing from "… less than 100  ms" to "… less than 1 ms".

**Line 134 and Fig.1: It would be informative to give the height and some other relevant parameters used in calculating the results shown in Fig. 1, presuming they are relevant to the present modelling.**
The focus of figure 1 is to illustrate the implementation of the electron temperature calculations during heating. However, as the referee suggests, it would be informative to mention the parameters used to calculate the results in figure 1. We will add the relevant parameters to the manuscript, both at line 134 and in the figure caption.

We have changed the text in line 134 from:
 "In Fig. 1 we show $Q - L$ as a function of $T_e$ with $I_0$, where the zero-point is illustrated as a blue coloured star. Also in Fig. 1 we show the changed intensity, illustrated as $I_1$ and the zero-point for $Q - L$ with $I_1$ is marked as a magenta coloured star."
to (now in line 138):
"Figure 1 shows $Q - L$ as a function of $T_e$. This figure illustrate that loss due to absorption can change the location of the zero-point of $Q - L$. We have used the following parameters to calculate $Q - L$: Height 75 km, ionospheric night conditions, model run with the presence of MSP, frequency 5 MHz and power 700 MW. Figure 1 shows the zero-point of $Q - L$ with $I_0$, illustrated as a blue-coloured star and the zero-point of $Q - L$ with the changed intensity $I_1$, illustrated as a magenta-coloured star." We have added the following text to the figure caption for figure 1: "We have used the following parameters to calculate $Q - L$: Height 75 km, ionospheric night conditions, model run with the presence of MSP, frequency 5 MHz and power 700 MW."

**Figure 2: This is fine, but it is not indicated that the 'Modelling during heating" is performed at each height from below before incrementing to the next height. This is well described in the text. I suggest that at least a sentence is added in the figure caption describing this.**
It is true that we do not mention this in the figure nor in the figure caption. We will do as the referee suggests and add a sentence to the figure caption describing that the modelling during heating is performed at each height from below before going to the next height. We have added the following text to the figure caption of figure 2: "We perform the modelling during heating at each height from below before going to the next height, moving upward from the initial height to the final height."

**Line 175:  The acronymn "SIC" is mentioned here for the first time. It should be explained what it stands for and a reference should be given.**
The referee is right; we have forgotten to explain what SIC stands for and forgotten to give a reference. We will add a reference to the SIC model and write out the acronym. Now the sentence reads: "For the initial conditions, the following parameters are taken from the Sodankylä Ion Chemistry (SIC) model (Turunen et al., 1996)."

**Section 3: Although the details of the MSP model are given in the references Baumann et al. 2013 and Megner et al. 2006, it would be useful for the reader without a detailed knowledge of MSP to have a brief, probably simplified description of the MSP height distribution without having to look up these references. Is it an idealised model or from measurements? For example are the MSP fairly uniformly distributed over the height ranges mentioned or are they in thin layers?**

This is a good suggestion from the referee. We have added a brief description at line 173 (section 3) of the MSP height distribution to the manuscript: "Megner et al. 2006 calculates the MSP number density profile by using a one-dimensional model, where the MSP height distribution varies with size. The number density of smaller MSPs (less than 15 nm) increases with altitude, while larger sizes are more abundant at lower altitudes between 60-70 km. Overall the number density of MSPs increases from 60 km to a maximum at around 80-83 km, and then decreases above. For an overview of the different MSP number density profiles, we refer the reader to figure A1 in the appendix."

**Figure 3: The red shaded box the reaction from P to Pp would seem to be photo-detachment so should the wiggly arrow labelled h(nu2) not be pointing away from P instead of towards P?**

The wiggly arrow indicates the incoming solar photon that detaches an electron from the surface of a neutral or negatively charged MSP. We have added the following to the figure caption of figure 3: "….where the wiggly arrow indicates the incoming solar photon that detaches an electron from the surface of a neutral or negatively charged MSP."

**Line 194-195: This sentence is a repeat of the sentence in line 193 but with a wrong "five cases" instead of four cases. Delete it.**

The referee is correct; line 194-195 is a repeat of the sentence in line 193. We will remove it.

**Section 5, in particular line 220: In discussing the 'open question' of the discrepancy between calculated and modelled electron temperature enhancements, a suggested explanation put forward in a later paper by Senior et al. should be mentioned, namely that the ERP of the heater may be overestimated because of the assumption of a perfectly conducting ground under the heating antennas is probably not met. This is discussed in section 6.6 and the conclusions of the paper 'Measurements and Modelling of Cosmic Noise Absorption Changes due to Radio Heating of the D-Region Ionosphere', Senior, A., M.T. Rietveld, F. Honary, W. Singer, M. J. Kosch, J. Geophys. Res., 116, A04310, doi:10.1029/2010JA016189, 2011.**

As suggested by the referee, we will mention the explanation put forward by Senior et al. 2011. At line 266, we have added a paragraph: "An explanation for the discrepancy between models and observations suggested by Senior et al. 2011 is that the modeled heater ERP is lower than predicted because the assumption of a perfect reflecting ground around the antenna might not be applicable. Senior et al. 2011 found that the overestimation is reduced when modelling the ERP with more realistic ground assumptions."

**Appendix, line 281: Does atomic oxygen really play an important role at these heights since it is a minor constituent here? This is discussed in section 6.3 of Senior et al. 2010.**

As mentioned by Senior et al. 2010, the concentration of atomic oxygen is very small at the relevant height region of 60-100 km. At theses heights, atomic oxygen is less important compared to the other species, i.e. molecular oxygen and molecular nitrogen. However, above 100 km, the concentration of atomic oxygen increases. At 120 km, the concentration of atomic oxygen is in the same order of magnitude as the concentration of molecular oxygen. We choose to include atomic oxygen since our modelling is from 60-120 km. The phrasing that atomic oxygen plays an important role is incorrect; therefore, we will change the sentence in line 281 from:

"In addition, atomic oxygen [O] plays an important role through the impact excitation of fine structure levels of its ground state (see Pavlov and Berrington 1999 references given there)."

to (now line 349):

"Even though the concentration of atomic oxygen is very small between 60-100 km (as discussed by Senior et al. 2010), we will include electron cooling rates for atomic oxygen [O] through the impact excitation of fine structure levels of its ground state (see Pavlov and Berrington 1999 references given there). We do this because our modelling is between 60-120 km, and the concentration of atomic oxygen increases above 100 km. At 120 km, the concentration of atomic oxygen is in the same order of magnitude as the concentration of molecular oxygen."

**Equation A6: One bracket is not closed.**
It is true that equation A6 is not closed. We have removed the excess bracket to the left. Now the equation is identical to the equation in the reference paper by Pavlov and Berrington 1999.

**Reference:**
**Stubbe, P., H. Kopka, M. T. Rietveld, R. L. Dowden, J. Atmos. Terr. Phys, 44, 12, 1123-1135, 1982, ELF and VLF wave generation by modulated heating of the current carrying lower ionosphere.**

**Technical errors**
* * *
**In the attached .pdf text I have marked spelling and grammar faults by highlighting in yellow without specifying the error except in some cases listed below. Many of the grammar faults are typical for Scandinavian writers: conjugation of verbs for singular or plural (adding or missing an 's' on the verb). In other cases plural nouns are missing the 's'.**
We have looked through the attached .pdf text and fixed the spelling errors and grammar faults.

**Lines 109 and 280:  Should be "lose".**
We have now changed it to "lose".

**Line 210: "conditions".**
We have changed it to "conditions".
* * *
In addition, we have made a number of modifications suggested by the other referee.
* * *
In line 341 (appendix), we have added the following regarding the MSP number density profiles:
"Figure A1 shows the different MSP number density profiles: panel a) shows the MSP autumn case (Megner et al. 2006) for 8. September, panel b) shows the MSP winter case (Megner et al. 2008) for 1. January and panel c) shows the MSP summer case (Megner et al. 2008) for 20. July. The MSP number density profile for autumn and winter is quite similar. However, the difference between the winter and summer case is quite significant, particularly for the larger sizes above 5 nm, which is more abundant for the summer case and extends to a higher altitude as well."

[Figure]

*Figure A1: Different MSP number density profiles for a) the autumn case, b) the winter case and c) the summer case. The autumn case is from Megner et al. 2006, while the winter and summer case is from Megner et al. 2008.*